# STRUCTURE-AWARE BIPARTITE REPRESENTATIONS FOR EFFICIENT MILP BRANCHING

## ABSTRACT

Efficient branching variable selection is pivotal to the performance of Branch-and-Bound (B&B) algorithms in Mixed Integer Linear Programming (MILP). Despite advances in traditional heuristics and graph-based learning methods, these approaches often fail to exploit the latent block structures inherent in many MILP problems. To address this limitation, we propose a novel graph representation that incorporates explicit block-structure annotations. By classifying variables and constraints according to their roles in block decompositions and augmenting edges with block identifiers, our method enables MILP solvers to better recognize localized patterns and global couplings. Through extensive experiments on six diverse MILP benchmarks, we demonstrate that our approach significantly improves upon state-of-the-art graph neural network baselines. Specifically, our method reduces search tree sizes by 2%–4% on standard instances and by 11%–13% on transfer instances, while decreasing solver runtime by 6%–6.66% on standard instances and by 5.5%–6% on transfer instances. Notably, these improvements are achieved without compromising solution quality. Our work highlights the importance of integrating structural priors into combinatorial optimization frameworks.

## 1 INTRODUCTION

Mixed Integer Linear Programming (MILP) is a fundamental optimization technique with extensive applications in logistics, scheduling, network design, and resource allocation (Pochet & Wolsey, 2006; Wu et al., 2013; Sawik, 2011; Malandraki & Daskin, 1992). The Branch-and-Bound (B&B) algorithm (Land & Doig, 2009) is pivotal for solving MILP problems, with its efficiency heavily dependent on the selection of the branching variable at each node (Zhang et al., 2023). Traditional branching heuristics, including strong branching (SB) (Applegate et al., 1995), pseudo-cost branching (PC) (Bénichou et al., 1971), and reliability branching (RB) (Achterberg et al., 2005), are general-purpose strategies that perform consistently across various MILP problems. While SB ensures high solution quality through LP-relaxation lookahead, PC approximates SB's effectiveness at lower computational costs by leveraging historical bound-improvement statistics. RB further bridges SB and PC by dynamically switching between exact lookahead and stabilized pseudocosts. Despite their widespread applicability and effectiveness, these conventional heuristics operate uniformly across different MILP instances and fail to account for latent structural patterns that distinguish problem families.

Early approaches to MILP solving relied predominantly on expert-designed heuristics, which did not account for *instance-specific* structural characteristics. To address this limitation, machine learning-based methods were introduced, using fixed-length, manually engineered feature vectors to represent static and dynamic solver states. These features included objective coefficients, reduced costs, pseudocosts, LP slackness, infeasibility measures, and branching history Marcos Alvarez et al. (2014); Khalil et al. (2016). While these methods achieved reduced search tree sizes and runtime improvements in specific settings, their reliance on hand-crafted descriptors and instance-specific tuning limited their generalizability across problem sizes and structures.

Recent advancements in graph-based learning methods have introduced new opportunities for enhancing MILP solvers. Gasse et al. (2019) pioneered a bipartite graph representation for MILPs, modeling variables and constraints as nodes connected by coefficient-weighted edges, and applied Graph Neural Networks (GNNs) to predict branching scores. Subsequent advancements, such as

the incorporation of cut separators, graph attention mechanisms, and trajectory features (Nair et al., 2020; Zarpellon et al., 2021; Gupta et al., 2020; Seyfi et al., 2023; Parsonson et al., 2023), have further enhanced the adaptability of branching policies. However, these approaches remain insensitive to the *hidden block structures* that are prevalent in many MILP families. In addition, Chen et al. (2023) pointed out that GNNs applied to bipartite graph representations of MILPs have limited discriminative power and cannot distinguish some non-isomorphic graphs. They proposed augmenting the node features with random attributes to break symmetry, which suggests that certain latent problem information needs to be explicitly injected to effectively guide GNN learning. Mareček (2012) highlighted the existence of such block structures in MILP problems. Building on this, Liu et al. (2024) demonstrated that their MILP-StuDio framework leverages these block structures by reordering coefficient matrices to generate new problem instances. This approach not only preserves the structural and mathematical properties of the original problem but also improves the quality of the generated instances, leading to better performance in downstream learning tasks. Compared to the instance generation method proposed by Wang et al. (2023), MILP-StuDio delivers superior results, offering more effective and informative instances for solver training. However, neither MILP-StuDio nor existing GNN-based representations have yet integrated this block-structure information into the solver's decision-making process.

In this work, we address this critical gap by augmenting the bipartite graph representation with explicit structural encodings derived from *matrix block decomposition* (Liu et al., 2024). We extend the block decomposition method to detect block roles and classify variables into *master*, *block*, or *border* types, and constraints into *block*, *doubly-bordered*, or *master* categories. This extension enables the incorporation of additional structural information, thereby enhancing the model's ability to capture and utilize inherent patterns within MILP instances.

In summary, the contributions of this work are threefold:

1. We identify and address the limitations of current bipartite graph models in overlooking latent block patterns critical for solving MILP problems.

2. We propose and integrate explicit block-aware features into the bipartite graph representation, thereby improving structural modeling capabilities.

3. We empirically validate that our structure-enriched graph representations lead to more informed branching decisions across diverse MILP benchmarks, outperforming baseline GNN models that lack block-structure integration.

The remainder of this paper is structured as follows: Section 2 provides an overview of MILP solving techniques and advances in graph-based learning for branching. Section 3 elaborates on our proposed structure-aware augmented graph representation. Section 4 empirically evaluates the efficacy of our approach, and Section 5 summarizes our contributions.

## 2 PRELIMINARY

### 2.1 MILP FORMULATION

Mixed Integer Linear Programming (MILP) constitutes a fundamental class of optimization problems characterized by decision variables subject to both continuous and discrete (integer) constraints. The general formulation of a MILP is expressed as:

$$
\begin{aligned}
\text{minimize} \quad & c^T x \\
\text{subject to} \quad & Ax \leq b, \\
& x \in \mathbb{Z}^p \times \mathbb{R}^{n-p}
\end{aligned}
$$

In this formulation, $x \in \mathbb{Z}^p \times \mathbb{R}^{n-p}$ denotes the vector of decision variables, where $p$ variables are restricted to integer values and $n - p$ variables are continuous. The vector $c \in \mathbb{R}^n$ represents the coefficients of the linear objective function to be minimized. The matrix $A \in \mathbb{R}^{m \times n}$ and vector $b \in \mathbb{R}^m$ define the system of linear inequality constraints that the variables must satisfy. The goal is to identify the values of $x$ that minimize the objective function while adhering to the specified constraints.

## 2.2 BRANCHING IN MILP

MILP problems are inherently NP-hard. The predominant approach for solving them is the Branch-and-Bound (B&B) algorithm, which systematically explores the solution space by iteratively solving LP relaxations of the MILP. In the B&B framework, the integer constraints are relaxed to form a linear program (LP). If the LP relaxation yields an integer solution that satisfies all constraints, this solution is deemed valid. Otherwise, the algorithm partitions the feasible region by selecting a fractional decision variable $x_i$ and generating two subproblems:

$$x_i \leq \lfloor x_i^\star \rfloor \quad \text{or} \quad x_i \geq \lceil x_i^\star \rceil, \quad \exists i \leq p \mid x_i^\star \notin \mathbb{Z},$$

where $x_i^\star$ denotes the fractional value of $x_i$ in the current LP solution. This recursive partitioning constructs a search tree, with the algorithm terminating when the upper and lower bounds converge or further decomposition becomes infeasible, thereby establishing either the optimality or infeasibility of the problem.

A critical aspect of the B&B method is the selection of the branching variable. The choice of which variable to branch on can significantly influence the size of the search tree and the efficiency of the algorithm.

# 3 STRUCTURE-AWARE MILP BRANCHING

In real-world optimization, MILP instances often exhibit inherent *block structures*, which are prevalent in domains such as combinatorial auctions, facility location, item placement, multi-knapsack allocation, and workload balancing. These structures stem from the symmetries and repetitions in problem formulations, where multiple entities with similar attributes translate into variable and constraint groups with shared patterns. Thus, identifying and leveraging these structures is pivotal for designing efficient B&B techniques.

## 3.1 IDENTIFICATION OF BLOCK STRUCTURE

MILP instances typically manifest *block structures* within their Constraint-Coefficient Matrices (CCMs), characterized by repeated, sparsely populated submatrices (*block units*) that persist across problem instances, leading to block-wise nonzero distributions in CCMs. Key block structures include:

- **Block-Diagonal (BD)**: Independent subproblems with no coupling.

- **Bordered Block-Diagonal (BBD)**: Subproblems coupled through shared constraints.

- **Doubly Bordered Block-Diagonal (DBBD)**: Subproblems coupled via both shared variables and constraints.

These structures can be schematically represented as follows:

$$
\begin{pmatrix}
D_1 & & & \\
& D_2 & & \\
& & \ddots & \\
& & & D_k
\end{pmatrix}
\qquad
\begin{pmatrix}
D_1 & & & \\
& D_2 & & \\
& & \ddots & \\
& & & D_k \\
B_1 & B_2 & \cdots & B_k
\end{pmatrix}
\qquad
\begin{pmatrix}
D_1 & & & & F_1 \\
& D_2 & & & F_2 \\
& & \ddots & & \vdots \\
& & & D_k & F_k \\
B_1 & B_2 & \cdots & B_k & C
\end{pmatrix}
$$

(a) Block-diagonal     (b) Bordered block-diagonal     (c) Doubly bordered block-diagonal

$$(1)$$

In this setting, $D_i$ represents local block constraints, $B_i$ model coupling constraints across blocks, and $C$ represents global master constraints. Variables can be classified as *block variables*, *border variables* or *master variables*. Constraints are similarly categorized as *block constraints* (B-Cons), *master constraints* (M-Cons), and *doubly bordered constraints* (DB-Cons), enabling more fine-grained structural analysis. For instance, a MILP with a DBBD structure can be formulated as follows:

$$
\begin{aligned}
\min_{x \in \mathbb{Z}^p \times \mathbb{R}^{n-p}} \quad & c_1^\top x_1 + c_2^\top x_2 + \cdots + c_k^\top x_k + c_{k+1}^\top x_{k+1}, \\
\text{s.t.} \quad & D_i\, x_i + F_i\, x_{k+1} \;\leq\; b_i, \quad i = 1, 2, \ldots, k \\
& \big(\text{B-Cons if } F_i = 0, \text{ DB-Cons otherwise}\big), \\
& \sum_{i=1}^{k} B_i\, x_i + C\, x_{k+1} \;\leq\; b_{k+1}, \quad (\text{M-Cons}), \\
& \ell \;\leq\; x \;\leq\; u,
\end{aligned}
\tag{2}
$$

The identification of such structures typically involves reordering the rows and columns of CCMs to cluster nonzero entries. Specifically, given a MILP instance, the initial ordering of rows and columns in its CCMs follows the enumeration of constraints and variables as defined during problem instantiation. In this raw arrangement, potential block structures are often obscured by the arbitrary positioning of nonzero entries. To systematically unveil and exploit these latent patterns, we employ a structure detection mechanism provided by the Generic Column Generation (GCG) solver Gamrath & Lübbecke (2010). This detector computes suitable permutations of rows and columns that aggregate nonzero coefficients into coherent clusters. By reordering according to these permutations, distinct block structures emerge clearly within the CCM, thereby facilitating subsequent decomposition and solution strategies.

## 3.2 CLASSIFICATION OF CONSTRAINTS AND VARIABLES

After reordering the rows and columns of the CCMs to reveal their latent block structures, we systematically classify the rows and columns based on their roles within the decomposed matrix. This classification leverages statistical descriptors of nonzero coefficients to identify inherent patterns in the problem structure.

**Constraint Classification**:

- **Block constraints (B-Cons):** Constraints that lie entirely within a diagonal block $D_i$ and do not involve any border variables $x_{k+1}$. These constraints represent localized relationships within individual blocks.

- **Doubly bordered constraints (DB-Cons):** Constraints of the form $D_i x_i + F_i x_{k+1} \leq b_i$ where $F_i \neq 0$. These constraints couple local variables $x_i$ with border variables $x_{k+1}$, introducing interdependencies across blocks.

- **Master constraints (M-Cons):** The set of constraints $\sum_{i=1}^{k} B_i x_i + C x_{k+1} \leq b_{k+1}$, which involve only the border variables $x_{k+1}$. These constraints enforce global coherence across blocks, ensuring overall feasibility.

**Variable Classification**:

- **Master variables (Mt-Vars):** Variables $x_{k+1}$ that participate exclusively in the master block $C$, coordinating decisions across blocks.

- **Block variables (Bl-Vars):** Variables $x_i$ confined to a single diagonal block $D_i$, representing localized decisions within their respective blocks.

- **Border variables (Bd-Vars):** Variables $x_{k+1}$ that appear in both coupling matrices $F_i$ and the master block $C$, acting as bridges between local block decisions and the global problem structure.

This classification forms the foundation for augmenting the bipartite graph representation with structural role annotations, thereby enabling the solver to better exploit the problem's block-coupled nature during branching. Some of the visualization results are provided in Appendix A.1.8.

## 3.3 AUGMENTATION OF BIPARTITE GRAPH REPRESENTATION

We propose an augmented bipartite graph representation for MILP that explicitly encodes structural features derived from block decomposition based on Gasse et al. (2019). The following augmenta-

tion strategy is applied to the bipartite graph $\mathcal{G} = (V_x \cup V_C, E)$, where $V_x$ denotes variable nodes and $V_C$ denotes constraint nodes. Edges $(c_i, x_j) \in E$ represent non-zero coefficients $A_{ij}$ in the constraint matrix.

**Variable-Node Features (3-dimensional one-hot encoding).** For each variable node $x \in V_x$, we append a 3-dimensional one-hot vector $\mathbf{r}_x \in \{0, 1\}^3$ to encode its structural role within the MILP block decomposition:

$$\mathbf{r}_x = \begin{cases} [1, 0, 0], & \text{if } x \text{ is a master variable (Mt-Var),} \\ [0, 1, 0], & \text{if } x \text{ is a block variable (Bl-Var),} \\ [0, 0, 1], & \text{if } x \text{ is a border variable (Bd-Var).} \end{cases}$$

This encoding distinguishes variables based on their participation in master, block, or coupling components of the problem, enabling the model to leverage structural priors during optimization.

**Constraint-Node Features (3-dimensional one-hot encoding).** For each constraint node $c \in V_C$, we append a 3-dimensional one-hot vector $\mathbf{r}_c \in \{0, 1\}^3$ to indicate its structural classification:

$$\mathbf{r}_c = \begin{cases} [1, 0, 0], & \text{if } c \text{ is a block constraint (B-Cons),} \\ [0, 1, 0], & \text{if } c \text{ is a doubly bordered constraint (DB-Cons),} \\ [0, 0, 1], & \text{if } c \text{ is a master constraint (M-Cons).} \end{cases}$$

This feature distinguishes constraints based on their involvement in local blocks, global coupling, or master-level aggregations, thereby enriching the graph representation with critical structural patterns.

**Edge Features (1-dimensional normalized block ID).** For each edge $e = (c_i, x_j)$, we compute a scalar feature $b_e \in [0, 1]$ to encode block affiliation:

$$b_e = \frac{\ell}{k},$$

where $\ell$ is the index of the diagonal block $D_\ell$ (or the master block if $c_i$ is an M-Cons constraint) and $k$ is the total number of blocks. Edges sharing the same $b_e$ value belong to the same block unit, allowing the model to identify intra-block relationships and inter-block couplings.

**Integration with Downstream Models.** The proposed structural features ($\mathbf{r}_x$, $\mathbf{r}_c$, and $b_e$) are concatenated with existing node and edge attributes, such as coefficient magnitudes and LP statistics, to form the final input vectors for graph-based learning models. This augmented representation enables downstream models (e.g., Graph Neural Networks) to better capture localized block patterns and global structural dependencies. By explicitly encoding each node's role, block membership, and each edge's block affiliation, the enhanced bipartite graph captures both local block patterns and global coupling structure, improving performance on tasks such as learned branching and cut selection. A detailed analysis of additional features and their impact can be found in the Appendix A.2 and A.5.

### 3.4 IMITATION LEARNING PARADIGM FOR BRANCHING DECISION

After enriching the standard bipartite graph representation with explicit structural annotations, we adopt the experimental protocol of Gasse et al. (2019). Specifically, we employ imitation learning to replicate the Strong Branching (SB) strategy of the MILP solver. Imitation learning is particularly well-suited for this task due to its ability to leverage expert demonstrations while maintaining computational efficiency in large-scale optimization problems. This approach allows us to systematically evaluate how our augmented graph features influence branching decisions. The overall architecture of the proposed framework is illustrated in Figure 1.

Formally, let $\mathcal{D} = \{(s_i, a_i)\}_{i=1}^N$ denote a dataset of expert demonstrations, where $s_i$ represents the state at decision $i$, and $a_i$ represents the action taken at that decision. The objective is to learn a policy $\pi_\theta(a|s)$ parameterized by $\theta$ that minimizes the discrepancy between the agent's actions and

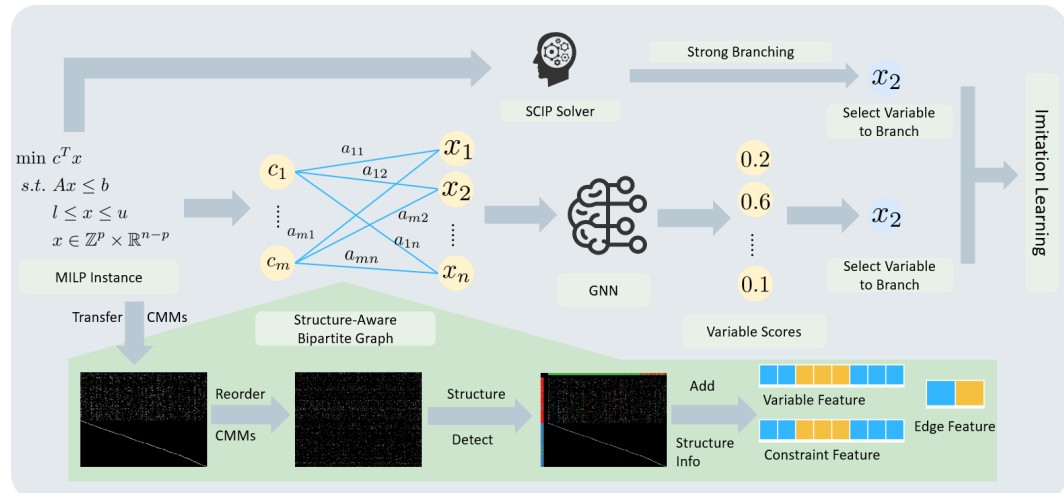

Figure 1: Structure-aware Bipartite Representation Imitation Learning Framework.

the expert's actions. This discrepancy is quantified using the negative log-likelihood loss:

$$\mathcal{L}(\theta) = -\sum_{i=1}^{N} \log \pi_\theta(a_i|s_i).$$

Training involves optimizing $\theta$ to minimize $\mathcal{L}(\theta)$, enabling the agent to replicate the expert's branching decisions.

## 4 EXPERIMENTS

### 4.1 EXPERIMENTAL SETTINGS

We evaluate the effectiveness of our proposed method on a diverse collection of benchmark MILP datasets widely used in prior work. The goal is to assess whether incorporating structural information into the graph-based variable selection strategy leads to improved solver efficiency across problem families of varying complexity.

**Datasets.** We consider six MILP benchmark datasets: Set Covering (SC) (Gasse et al., 2019), Combinatorial Auctions (CA) (Leyton-Brown et al., 2000), Capacitated Facility Location (CFL) (Cornuéjols et al., 1991), Maximum Independent Set (MIS) (Bergman et al., 2016; Gasse et al., 2019), Multiple Knapsack (MK) (Gasse et al., 2019), Item Placement (IP) (Gasse et al., 2022) and Load Balancing(LB) (Gasse et al., 2022). These datasets encompass a range of structural patterns, including both structured and unstructured instances, offering a comprehensive testbed for branching strategies. Each benchmark is partitioned into Standard and Transfer instances according to solver-determined difficulty. More details can be found in Appendix A.1.

**Environmental Setup.** All experiments are conducted using the SCIP 8.1.0 optimization suite (Bestuzheva et al., 2021), integrated with the Ecole library (Prouvost et al., 2020), leveraging its capabilities for machine learning in combinatorial optimization. The computational infrastructure comprises eight NVIDIA TITAN Xp GPUs, each equipped with 16 GB of memory. A strict time limit of 3600 seconds is imposed on the solver for both training data generation and instance resolution to maintain consistency and feasibility across evaluations. To isolate the impact of our proposed method and ensure a fair comparison, we disable all advanced heuristic and decomposition techniques in SCIP, such as plane cuts, diving heuristics, and presolve routines, which might otherwise confound the assessment of our method's standalone performance.

**Baselines.** We compare our proposed method against several widely adopted baselines to ensure a fair and comprehensive evaluation. These baselines include:

Table 1: Comparative results in the solving time and the size of the search tree on the *standard testing instances*, which are of the same size as training instances. Only *neural methods* are compared in the number of nodes.

| | CA | MK | SC | MIS | CFA |
|---|---|---|---|---|---|
| Model | Time (s) ↓ | Time (s) ↓ | Time (s) ↓ | Time(s) ↓ | Time (s) ↓ |
| FSB | $19.52 \pm 10.52\%$ | $2.49 \pm 374.67\%$ | $3.78 \pm 8.10\%$ | $90.23 \pm 18.10\%$ | $106.53 \pm 13.44\%$ |
| PB | $4.27 \pm 20.64\%$ | $0.81 \pm 103.46\%$ | $0.68 \pm 12.68\%$ | $40.94 \pm 66.52\%$ | $56.52 \pm 14.53\%$ |
| RPB | $6.04 \pm 14.91\%$ | $0.68 \pm 73.61\%$ | $2.00 \pm 18.87\%$ | $11.93 \pm 20.59\%$ | $132.14 \pm 16.39\%$ |
| GNN | $4.49 \pm 12.24\%$ | $13.96 \pm 283.84\%$ | $\mathbf{1.12 \pm 9.93\%}$ | $11.71 \pm 22.49\%$ | $92.00 \pm 15.29\%$ |
| GNN_DEC | $\mathbf{4.21 \pm 11.51\%}$ | $12.10 \pm 359.44\%$ | $1.13 \pm 10.07\%$ | $11.05 \pm 22.35\%$ | $89.64 \pm 14.14\%$ |
| GNN_DEC2 | $4.40 \pm 11.68\%$ | $\mathbf{11.77 \pm 342.80\%}$ | $1.16 \pm 10.15\%$ | $\mathbf{10.15 \pm 21.31\%}$ | $\mathbf{87.00 \pm 14.53\%}$ |
| Model | # Nodes ↓ | # Nodes ↓ | # Nodes ↓ | # Nodes ↓ | # Nodes ↓ |
| FSB | $109.96 \pm 8.51\%$ | $413.49 \pm 4022.18\%$ | $11.27 \pm 3.26\%$ | $116.22 \pm 23.27\%$ | $221.05 \pm 12.60\%$ |
| PB | $1690.52 \pm 24.63\%$ | $290.02 \pm 887.14\%$ | $60.81 \pm 18.17\%$ | $18378.16 \pm 76.69\%$ | $363.60 \pm 10.56\%$ |
| RPB | $136.08 \pm 26.19\%$ | $225.58 \pm 250.15\%$ | $8.88 \pm 17.20\%$ | $410.26 \pm 45.23\%$ | $215.59 \pm 10.72\%$ |
| GNN | $327.83 \pm 14.98\%$ | $358.45 \pm 556.93\%$ | $\mathbf{38.26 \pm 7.02\%}$ | $553.69 \pm 24.78\%$ | $341.38 \pm 12.84\%$ |
| GNN_DEC | $\mathbf{307.83 \pm 14.32\%}$ | $\mathbf{327.25 \pm 987.61\%}$ | $38.36 \pm 7.56\%$ | $595.27 \pm 24.51\%$ | $334.90 \pm 10.46\%$ |
| GNN_DEC2 | $324.53 \pm 14.68\%$ | $330.40 \pm 779.40\%$ | $39.03 \pm 7.36\%$ | $\mathbf{552.28 \pm 24.08\%}$ | $\mathbf{332.66 \pm 11.31\%}$ |
| Model | # Gap ↓ | # Gap ↓ | # Gap ↓ | # Gap ↓ | # Gap ↓ |
| FSB | $0.000000 \pm 0.00\%$ | $0.000011 \pm 0.00\%$ | $0.000000 \pm 0.00\%$ | $0.000000 \pm 0.00\%$ | $0.000000 \pm 0.00\%$ |
| PB | $0.000000 \pm 0.00\%$ | $0.000000 \pm 0.00\%$ | $0.000000 \pm 0.00\%$ | $0.000000 \pm 0.00\%$ | $0.000000 \pm 0.00\%$ |
| RPB | $0.000000 \pm 0.00\%$ | $0.000000 \pm 0.00\%$ | $0.000000 \pm 0.00\%$ | $0.000000 \pm 0.00\%$ | $0.000000 \pm 0.00\%$ |
| GNN | $\mathbf{0.000000 \pm 0.00\%}$ | $0.000012 \pm 0.00\%$ | $\mathbf{0.000000 \pm 0.00\%}$ | $\mathbf{0.000000 \pm 0.00\%}$ | $\mathbf{0.000000 \pm 0.00\%}$ |
| GNN_DEC | $\mathbf{0.000000 \pm 0.00\%}$ | $0.000014 \pm 0.00\%$ | $\mathbf{0.000000 \pm 0.00\%}$ | $\mathbf{0.000000 \pm 0.00\%}$ | $\mathbf{0.000000 \pm 0.00\%}$ |
| GNN_DEC2 | $\mathbf{0.000000 \pm 0.00\%}$ | $\mathbf{0.000006 \pm 0.00\%}$ | $\mathbf{0.000000 \pm 0.00\%}$ | $\mathbf{0.000000 \pm 0.00\%}$ | $\mathbf{0.000000 \pm 0.00\%}$ |

1. **Full Strong Branching (FSB)**: A deterministic branching rule known for its high solution quality but significant computational overhead.

2. **Pseudocost Branching (PB)**: A heuristic method leveraging historical statistics to approximate the effectiveness of branching decisions.

3. **Reliability Pseudocost Branching (RPB)**: A hybrid approach combining features of FSB and PB for improved reliability (Achterberg et al., 2005).

4. **Graph Convolutional Neural Network (GCNN)**: A graph-based learning policy from prior work (Gasse et al., 2019) that serves as a strong machine learning baseline.

**Training Protocol.** For imitation learning, we collect training samples by solving the training instances using FSB under a 3600-second timeout. At each branching decision, we record the bipartite graph representation of the MILP state along with the FSB-selected variable. For each dataset, we collect 160k samples across training instances. Our models are trained using behavioral cloning with cross-entropy loss, and the policy is optimized using gradient descent.

**Evaluation Metrics.** We follow the evaluation method established in Gasse et al. (2019). Specifically, we report:

- **Time:** 1-shifted geometric mean of solving times across validation instances.
- **Node:** 10-shifted geometric mean of the number of branch-and-bound nodes.

All reported metrics represent the average over five independent runs using different random seeds during inference. For Item Placement (IP), we further report the *dual integral reward* (Gasse et al., 2022), defined as $R = \int_0^T z_t \, dt - T \cdot x$, which quantifies the area between the dual bound trajectory and the optimal objective over time. This metric provides a nuanced evaluation of early progress toward optimality.

## 4.2 RESULTS AND DISCUSSIONS

Tables 1 and 2 present the comparative results on standard and transfer testing instances, respectively. The baseline neural method, denoted as GNN, serves as a reference, while GNN_DEC and

Table 2: Comparative results in the solving time and the size of the search tree on the *transfer testing instances*, which are larger than the training instances. We bold the best results for each metric. Only *neural methods* are compared in the number of nodes.

| | CA | MK | SC | MIS | CFA |
|---|---|---|---|---|---|
| Model | Time (s) ↓ | Time (s) ↓ | Time (s) ↓ | Time(s) ↓ | Time (s) ↓ |
| FSB | 1759.60 ± 5.58% | 55.88 ± 204.10% | 89.58 ± 7.69% | 3265.70 ± 4.57% | 904.09 ± 13.93% |
| PB | 171.82 ± 24.83% | 12.76 ± 139.75% | 10.26 ± 14.86% | 2899.30 ± 30.37% | 472.48 ± 10.79% |
| RPB | 111.27 ± 9.84% | 19.46 ± 119.53% | 20.29 ± 11.89% | 161.38 ± 24.27% | 693.91 ± 9.14% |
| GNN | 140.56 ± 6.18% | 184.49 ± 187.76% | 14.72 ± 6.82% | 131.07 ± 14.73% | **422.59 ± 10.62%** |
| GNN_DEC | **122.16 ± 9.87%** | 112.04 ± 164.90% | **14.68 ± 5.22%** | 137.78 ± 13.47% | 509.50 ± 11.75% |
| GNN_DEC2 | 134.07 ± 7.86% | **93.59 ± 232.95%** | 14.87 ± 6.63% | **130.55 ± 12.45%** | 538.20 ± 12.63% |
| Model | # Nodes ↓ | # Nodes ↓ | # Nodes ↓ | # Nodes ↓ | # Nodes ↓ |
| FSB | 1485.87 ± 6.85% | 12985.34 ± 366.21% | 110.41 ± 4.62% | 789.19 ± 9.04% | 246.23 ± 9.48% |
| PB | 33336.27 ± 25.50% | 8896.75 ± 250.70% | 986.05 ± 17.00% | 559588.10 ± 43.00% | 560.89 ± 8.56% |
| RPB | 8508.56 ± 10.37% | 7945.67 ± 236.02% | 349.08 ± 17.11% | 12198.61 ± 19.47% | 302.79 ± 9.68% |
| GNN | 10117.72 ± 6.58% | 7608.66 ± 267.30% | 478.38 ± 6.52% | **6849.00 ± 12.97%** | 593.43 ± 8.49% |
| GNN_DEC | **8328.25 ± 11.18%** | 4694.16 ± 228.32% | **470.91 ± 5.20%** | 7980.83 ± 13.16% | 585.39 ± 10.36% |
| GNN_DEC2 | 9477.79 ± 9.31% | **3608.23 ± 414.54%** | 487.53 ± 7.34% | 7670.01 ± 14.28% | **570.76 ± 8.57%** |
| Model | # Gap ↓ | # Gap ↓ | # Gap ↓ | # Gap ↓ | # Gap ↓ |
| FSB | 0.003547 ± 0.01% | 0.000039 ± 0.00% | 0.000000 ± 0.00% | 0.034084 ± 0.02% | 0.000013 ± 0.00% |
| PB | 0.000000 ± 0.00% | 0.000019 ± 0.00% | 0.000000 ± 0.00% | 0.026953 ± 0.01% | 0.000000 ± 0.00% |
| RPB | 0.000000 ± 0.00% | 0.000000 ± 0.00% | 0.000000 ± 0.00% | 0.000000 ± 0.00% | 0.000000 ± 0.00% |
| GNN | **0.000000 ± 0.00%** | 0.000168 ± 0.00% | **0.000000 ± 0.00%** | **0.000000 ± 0.00%** | **0.000000 ± 0.00%** |
| GNN_DEC | **0.000000 ± 0.00%** | **0.000000 ± 0.00%** | **0.000000 ± 0.00%** | **0.000000 ± 0.00%** | **0.000000 ± 0.00%** |
| GNN_DEC2 | **0.000000 ± 0.00%** | **0.000000 ± 0.00%** | **0.000000 ± 0.00%** | **0.000000 ± 0.00%** | **0.000000 ± 0.00%** |

Table 3: Comparative results in the dual integral reward on the *Item Placements*. We bold the best results for each metric.

| Time Limit | 60s | 120s | 240s | 480s | 900s |
|---|---|---|---|---|---|
| Model | # Dual Integral ↑ | # Dual Integral ↑ | # Dual Integral ↑ | # Dual Integral ↑ | # Dual Integral ↑ |
| RPB | 186.67 | 377.04 | 770.06 | 1581.29 | 3070.54 |
| GNN | 193.57 | 392.33 | 796.78 | 1622.29 | 3108.10 |
| GNN_DEC | **193.95** | **401.65** | **820.39** | **1687.20** | **3316.00** |
| GNN_DEC2 | 190.29 | 388.32 | 787.89 | 1591.40 | 3063.49 |

GNN_DEC2 correspond to our proposed models that incorporate explicit block-structure information and block decomposition, respectively.

Table 1 reports the comparative performance of GNN-based branching policies against classical branching methods on the standard testing instances, which are of the same size as the training instances. All GNN-based models achieve competitive solve times. In particular, GNN_DEC obtains the fastest average solve time on the CA problem family (4.21 s), whereas GNN_DEC2 attains the best performance on MK (11.77 s), MIS (10.15 s), and CFA (87.00 s). For SC, the standard GNN achieves the lowest solve time (1.12 s). In terms of search tree size, GNN_DEC produces the smallest trees on CA (308 nodes) and MK (327 nodes), while GNN_DEC2 achieves minimal node counts on MIS (333 nodes) and CFA (332 nodes). These results demonstrate that leveraging decomposition-based enhancements in the bipartite graph representation allows GNN policies to improve both solving efficiency and search tree reduction.

Table 2 presents the results on larger transfer testing instances. Here, GNN-based policies consistently outperform strong branching (FSB) and pseudo-cost branching (PB) in terms of solve time. Specifically, GNN_DEC achieves the fastest average solve time on CA (122.16 s) and SC (14.68 s), whereas GNN_DEC2 attains the best performance on MK (93.59 s) and MIS (130.55 s). Regarding the search tree size, GNN_DEC produces the smallest trees on CA (8,328 nodes) and SC (471 nodes), while GNN_DEC2 yields minimal node counts on MK (3,608 nodes) and MIS (571 nodes). These observations indicate that decomposition-aware GNN policies generalize effectively to larger instances, consistently reducing both solve time and search tree size relative to traditional branching methods.

Table 4: Comparative results in the dual integral reward on the *Load Balancing*. We bold the best results for each metric.

| Time Limit | 60s | 120s | 240s | 480s | 900s |
|---|---|---|---|---|---|
| Model | # Dual Integral ↑ | # Dual Integral ↑ | # Dual Integral ↑ | # Dual Integral ↑ | # Dual Integral ↑ |
| RPB | **41992.30** | **83950.14** | 167826.25 | 335581.62 | 629178.86 |
| GNN | 41952.76 | 83911.39 | **167890.70** | **335932.05** | 630241.37 |
| GNN_DEC | 41950.14 | 83903.59 | 167859.67 | 335924.58 | **630293.12** |
| GNN_DEC2 | 41949.26 | 83901.66 | 167856.51 | 335898.38 | 630237.44 |

Table 3 presents the dual integral reward on the Item Placement instances under different time budgets. All GNN-based policies outperform classical baselines, with GNN_DEC achieving the highest reward across all time limits, ranging from 209.91 at 60 s up to 3,469.19 at 900 s. This demonstrates that incorporating block-structure annotations into GNN policies enhances their anytime performance, enabling progressive improvement of dual bounds over the course of the computation.

Table 4 shows the dual integral reward on the Load Balancing instances under varying time budgets. All GNN-based policies achieve competitive performance relative to pseudo-cost branching (RPB). At shorter time limits (60 s and 120 s), RPB slightly outperforms the GNN variants, achieving rewards of 41,992.30 and 83,950.14, respectively. However, as the time budget increases, GNN-based policies progressively surpass RPB. Specifically, GNN attains the highest rewards at 240 s (167,890.70) and 480 s (335,932.05), while GNN_DEC reaches the maximal reward at 900 s (630,293.12). These results confirm that decomposition-aware GNN policies provide strong anytime performance, effectively improving dual bounds over extended computation times across different problem domains.

It is worth noting that the CA, MK, CFA, IP, and LB families exhibit clear block-structured patterns in their Constraint-Coefficient Matrices (CCMs), whereas the SC and MIS problem families lack such distinct block structures. Consequently, the relative advantage of the DEC variants (GNN_DEC and GNN_DEC2) is most pronounced on structurally rich problem families, where they achieve the largest reductions in solve time and search tree size. On the less structured SC and MIS families, their performance improvements remain moderate. This observation is consistent with our hypothesis that explicit block-structure annotations enhance branching decisions most effectively when the underlying MILP problem exhibits inherent decomposable structures. For further details on the impact of feature design, additional results are presented in Appendix A.5.

## 5 CONCLUSION

This paper presents a novel methodology for enhancing the Branch-and-Bound (B&B) algorithm for Mixed Integer Linear Programming (MILP) by integrating block-structure information into the bipartite graph representation. Our experimental results highlight that MILP solvers can significantly benefit from leveraging structural priors such as block decomposition. By incorporating these priors into the bipartite graph representation, our approach achieves a better balance between local block patterns and global coupling structures. These findings underscore the importance of structural-aware representations for improving the efficiency of combinatorial optimization algorithms.

Despite these advances, our method has limitations. The effectiveness of the proposed technique is sensitive to the accuracy of block-structure detection and classification. Misclassifications can lead to suboptimal branching decisions and reduced solver performance. Furthermore, the current framework assumes static block structures, which may not hold in dynamic real-world problems where block structures can evolve. Future research could explore dynamic learning of block structures and test the approach on large-scale industrial MILP instances to further assess its generalizability and robustness.

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

# A APPENDIX

## A.1 DATASETS

We generate five classes of combinatorial optimization problems using the `Ecole` library (Prouvost et al., 2020). The considered problem types include **Set Cover (SC)**, **Capacitated Facility Location (CFL)**, **Combinatorial Auction (CA)**, **Independent Set (IS)**, and **Multiple Knapsack (MK)**. These problem instances are generated following the standard procedures provided in `Ecole`, ensuring reproducibility and consistency across experiments.

The detailed parameter settings for each problem generator are summarized in Table 5.

Table 5: Problem instance generation settings. Standard Test corresponds to the same scale as training data, while Transfer Test uses a larger scale to evaluate generalization.

| Problem | Standard Test Scale | Transfer Test Scale | Problem Size Parameters |
|---|---|---|---|
| Set Cover | $500 \times 1000$ | $1000 \times 1000$ | #Rows = Cover Sets, #Cols = Elements |
| Capacitated Facility Location | $n = 100, m = 100$ | $n = 200, m = 100$ | $n$: Clients, $m$: Facilities |
| Combinatorial Auction | $|\mathcal{B}| = 500, |\mathcal{I}| = 100$ | $|\mathcal{B}| = 1000, |\mathcal{I}| = 200$ | $\mathcal{B}$: Bidders, $\mathcal{I}$: Items |
| Independent Set | $|V| = 500$ | $|V| = 1000$ | $|V|$: Graph Nodes (Affinity=4) |
| Multiple Knapsack | $n = 100, K = 6$ | $n = 200, K = 12$ | $n$: Items, $K$: Knapsacks |

### A.1.1 COMBINATORIAL AUCTION

Given a set of $m$ items $I = \{1, \ldots, m\}$ and a set of $n$ bids $B = \{1, \ldots, n\}$, each bid $b \in B$ specifies a subset of items $S_b \subseteq I$ and offers a price $v_b$. The goal is to select a collection of non-overlapping bids to maximize total revenue:

$$\max \quad \sum_{b=1}^{n} v_b x_b$$

$$\text{s.t.} \quad \sum_{b:\, i \in S_b} x_b \leq 1, \quad \forall i \in I,$$

$$x_b \in \{0, 1\}, \quad \forall b = 1, \ldots, n,$$

where $x_b = 1$ if bid $b$ is accepted and $0$ otherwise.

### A.1.2 SET COVER

Given a ground set of $m$ elements $U = \{1, \ldots, m\}$ and a family of $n$ subsets $S_1, S_2, \ldots, S_n \subseteq U$, each subset $S_j$ has an associated non-negative cost $c_j$. The goal is to select a minimum-cost collection of subsets that covers all elements of $U$. Formally:

$$\min \quad \sum_{j=1}^{n} c_j x_j$$

$$\text{s.t.} \quad \sum_{j:\, i \in S_j} x_j \geq 1, \quad \forall i \in U,$$

$$x_j \in \{0, 1\}, \quad \forall j = 1, \ldots, n,$$

where $x_j = 1$ if subset $S_j$ is selected and $0$ otherwise.

### A.1.3 CAPACITATED FACILITY LOCATION

Given a set of $m$ facilities $F = \{1, \ldots, m\}$ and a set of $n$ clients $C = \{1, \ldots, n\}$, each facility $i$ has a fixed opening cost $f_i$ and a capacity $s_i$. Serving client $j$ from facility $i$ incurs a unit transportation cost $c_{ij}$, and client $j$ has a demand $d_j$. The problem is to decide which facilities to open and how to

allocate client demands to minimize total cost:

$$\min \quad \sum_{i=1}^{m}\sum_{j=1}^{n} c_{ij}x_{ij} + \sum_{i=1}^{m} f_i y_i$$

$$\text{s.t.} \quad \sum_{j=1}^{n} d_j x_{ij} \leq s_i y_i, \qquad \forall i = 1, \ldots, m,$$

$$\sum_{i=1}^{m} x_{ij} = 1, \qquad \forall j = 1, \ldots, n,$$

$$x_{ij} \in \{0,1\}, \qquad \forall i = 1, \ldots, m, \ j = 1, \ldots, n,$$

$$y_i \in \{0,1\}, \qquad \forall i = 1, \ldots, m,$$

where $x_{ij} = 1$ indicates that client $j$ is assigned to facility $i$, and $y_i = 1$ indicates that facility $i$ is opened.

### A.1.4 INDEPENDENT SET

Given an undirected graph $G = (V, E)$ with $|V| = n$ vertices, each vertex $v \in V$ has a non-negative weight $w_v$. The independent set problem seeks a subset of vertices with maximum total weight such that no two adjacent vertices are both selected:

$$\max \quad \sum_{v \in V} w_v x_v$$

$$\text{s.t.} \quad x_u + x_v \leq 1, \quad \forall (u, v) \in E,$$

$$x_v \in \{0,1\}, \quad \forall v \in V,$$

where $x_v = 1$ indicates that vertex $v$ is included in the independent set.

### A.1.5 MULTIPLE KNAPSACK

Given $n$ items with respective prices $p_j$ and weights $w_j$ for $j = 1, \ldots, n$, and $m$ knapsacks with capacities $c_i$ for $i = 1, \ldots, m$, the multiple knapsack problem aims to place items into the knapsacks to maximize the total price of selected items while ensuring that the total weight in each knapsack does not exceed its capacity:

$$\max \quad \sum_{i=1}^{m}\sum_{j=1}^{n} p_j x_{ij}$$

$$\text{s.t.} \quad \sum_{j=1}^{n} w_j x_{ij} \leq c_i, \quad \forall i = 1, \ldots, m,$$

$$\sum_{i=1}^{m} x_{ij} \leq 1, \qquad \forall j = 1, \ldots, n,$$

$$x_{ij} \in \{0,1\}, \qquad \forall i = 1, \ldots, m, \ j = 1, \ldots, n,$$

where $x_{ij} = 1$ if item $j$ is placed into knapsack $i$, and 0 otherwise.

Additionally, we use two MILP benchmark problems from the ML4CO competition (Gasse et al., 2022):

### A.1.6 BALANCED ITEM PLACEMENT

This problem involves distributing items (e.g., files or processes) across containers (e.g., disks or machines) in a balanced manner. Each item may have multiple copies, but at most one copy can be placed in a single bin. The number of items that can be moved is limited, reflecting the practical scenario of a live system with an existing placement. Each instance is formulated as a multi-dimensional multi-knapsack MILP. The dataset contains 11,000 instances, pre-split into 9,900 training, 100 validation instances and 100 testing instances.

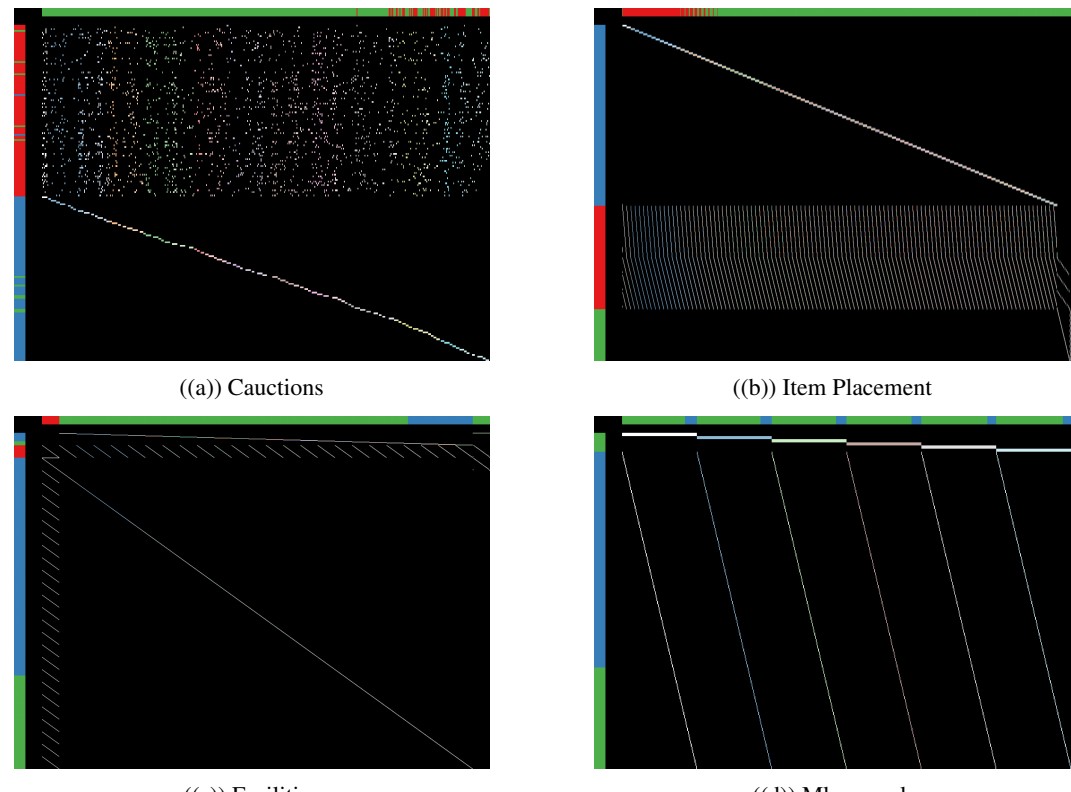

((a)) Cauctions        ((b)) Item Placement

((c)) Facilities        ((d)) Mknapsack

Figure 2: CCMs for four canonical benchmark instances, highlighting the characteristic block-structured patterns commonly encountered in mixed-integer linear programs.

### A.1.7 WORKLOAD APPORTIONMENT

This problem focuses on allocating workloads (e.g., data streams) across as few workers (e.g., servers) as possible, ensuring robustness against the failure of any single worker. Each instance is formulated as a bin-packing MILP with apportionment constraints. The dataset contains 11,000 instances, pre-split into 9,900 training, 100 validation instances and 100 testing instances.

### A.1.8 CCMS VISUALIZATION RESULT

The Constraint-Coefficient Matrices (CCMs) in MILPs encode the relationships between decision variables and constraints. These matrices often exhibit structured patterns that reflect the problem's inherent properties. Figure 2 visualizes the CCMs of four representative benchmark instances, revealing distinct structural characteristics across problem families.

For the combinatorial auction (CA), item placement (IP), and multiple knapsack (MK) instances, clear block-diagonal patterns emerge, where variables predominantly interact with localized subsets of constraints. Such patterns reflect the modular nature of these problems, where independent or weakly coupled subproblems contribute to the overall structure. In contrast, the capacitated facility location (FA) problem demonstrates a more uniform and dense CCM, lacking distinct block structures. This visual contrast aligns with the observed performance of decomposition-aware models: our proposed methods GNN_DEC and GNN_DEC2 demonstrate superior efficiency on block-structured instances, where the explicit modeling of localized variable-constraint interactions directly benefits branching decisions. Conversely, on more uniform and dense problems like FA, the advantages of decomposition-aware approaches diminish.

Table 6: Description of the constraint, edge and variable features in our bipartite state representation $s_t$

| Tensor | Index | Feature | Description |
|---|---|---|---|
| **C** | 0 | bias | Bias value, normalized with constraint coefficients. |
| | 1 | obj_cos_sim | Cosine similarity with objective. |
| | 2–4 | type | Constraint type (master, block, double block) as a one-hot encoding. |
| | 5 | is_tight | Tightness indicator in LP solution. |
| | 6 | dualsol_val | Dual solution value, normalized. |
| | 7 | age | LP age, normalized with the total number of LP iterations. |
| **E** | 0 | coef | Constraint coefficient, normalized per constraint. |
| | 1 | block id | (Constraint, Variable) pair belonging normalized block id. |
| **V** | 0 | coef | Objective coefficient, normalized. |
| | 1–4 | type | Variable type (binary, integer, implicit integer, continuous) as a one-hot encoding. |
| | 5–7 | block_info | Block or master variable indicator as a one-hot encoding. |
| | 8 | block_id | normalized Block id in decomposition. |
| | 9 | has_lb | Lower bound indicator. |
| | 10 | has_ub | Upper bound indicator. |
| | 11 | reduced_cost | Reduced cost, normalized. |
| | 12 | sol_val | Solution value. |
| | 13 | sol_frac | Solution value fractionality. |
| | 14 | sol_is_at_lb | Solution value equals lower bound. |
| | 15 | sol_is_at_ub | Solution value equals upper bound. |
| | 16 | age | LP age, normalized with the total number of LP iterations. |
| | 17 | inc_val | Value in incumbent solution. |
| | 18 | avg_inc_val | Average value in incumbent solutions. |
| | 19–22 | basis_status | Simplex basis status (lower, basic, upper, zero) as a one-hot encoding. |

## A.2 Implementation Details of Bipartite Graph Representation

The bipartite graph representation serves as a cornerstone of our model architecture, bridging the structural dependencies in MILP problems with graph-based learning techniques. In this work, we extend the bipartite graph framework initially proposed by Gasse et al. (2019) by augmenting it with richer structural annotations derived from block decomposition.

Specifically, the input features used in our model are detailed in Table 6. For each node and edge in the bipartite graph, we incorporate domain-specific descriptors such as objective coefficients, constraint coefficients, and solution statistics. Additionally, we propose a novel encoding scheme for structural roles, where variables and constraints are classified based on their participation in master, block, or border components of the problem. Edge features are further augmented with normalized block identifiers to highlight intra-block relationships and inter-block couplings. This enriched representation not only preserves the mathematical integrity of the MILP formulation but also injects critical structural priors that improve the model's ability to reason about the problem's inherent decomposability.

## A.3 Model Architecture

### A.3.1 Graph Convolution Network

Our model closely follows the architecture of Gasse et al. (2019), with minor modifications to suit our problem setting. The input is a bipartite graph state representation $s_t = (\mathcal{G}, C, V, E)$, where $C$ is the set of constraint nodes, $V$ is the set of variable nodes, and $E$ is the set of edges linking them. Each node and edge is associated with its own feature vector (see Table 6).

A single graph convolution is performed using two interleaved *half-convolutions*: first, information flows from variables to constraints; then, it flows back from constraints to variables. Formally, for each $c_i \in C$ and $v_j \in V$,

$$c_i \leftarrow f_C\Big(c_i, \sum_{(i,j)\in E} g_C(c_i, v_j, e_{ij})\Big),$$

$$v_j \leftarrow f_V\Big(v_j, \sum_{(i,j)\in E} g_V(c_i, v_j, e_{ij})\Big),$$

(3)

where $f_C, f_V, g_C, g_V$ are two-layer MLPs with ReLU activations.

After convolution, each variable embedding $v_j$ contains information from its neighbors. The policy is obtained by discarding the constraint nodes and applying a two-layer MLP to the variable embeddings, followed by a masked softmax to produce probabilities over the candidate branching variables:

$$\pi(a_t \mid s_t) = \text{SoftmaxMask}\big(\text{MLP}(v_j)\big), \tag{4}$$

where the mask ensures that only non-fixed LP variables are considered.

### A.3.2 GRAPH ATTENTION NETWORK

In addition to standard graph convolutions, we also consider two graph attentional operator, named GAT and GATv2 as proposed in Veličković et al. (2018) and Brody et al. (2022) separately.

For GAT, given a node $i$ with neighbors $\mathcal{N}(i)$, the updated embedding is computed as

$$\mathbf{x}'_i = \sum_{j \in \mathcal{N}(i) \cup \{i\}} \alpha_{i,j} \mathbf{\Theta}_t \mathbf{x}_j, \tag{5}$$

where $\mathbf{\Theta}_t$ is a learnable linear transformation and $\alpha_{i,j}$ is the attention coefficient between nodes $i$ and $j$.

The attention coefficients are obtained via a shared self-attention mechanism:

$$\alpha_{i,j} = \frac{\exp\left(\text{LeakyReLU}(\mathbf{a}_s^\top \mathbf{\Theta}_s \mathbf{x}_i + \mathbf{a}_t^\top \mathbf{\Theta}_t \mathbf{x}_j)\right)}{\sum_{k \in \mathcal{N}(i) \cup \{i\}} \exp\left(\text{LeakyReLU}(\mathbf{a}_s^\top \mathbf{\Theta}_s \mathbf{x}_i + \mathbf{a}_t^\top \mathbf{\Theta}_t \mathbf{x}_k)\right)}, \tag{6}$$

where $\mathbf{a}_s, \mathbf{a}_t$ are learnable attention vectors, and $\mathbf{\Theta}_s$ is a learnable linear mapping.

For GATv2, the attention coefficients are obtained via a shared self-attention mechanism:

$$\alpha_{i,j} = \frac{\exp\left(\mathbf{a}^\top \text{LeakyReLU}\left(\mathbf{\Theta}_s \mathbf{x}_i + \mathbf{\Theta}_t \mathbf{x}_j\right)\right)}{\sum_{k \in \mathcal{N}(i) \cup \{i\}} \exp\left(\mathbf{a}^\top \text{LeakyReLU}\left(\mathbf{\Theta}_s \mathbf{x}_i + \mathbf{\Theta}_t \mathbf{x}_k\right)\right)}. \tag{7}$$

These operator allow the model to weight contributions from neighboring nodes differently, enabling it to focus on the most relevant neighbors when updating node embeddings. The overall update is fully differentiable and can be stacked for multiple attention layers, similar to the convolutional GNN described above.

### A.4 FEATURE ANALYSIS OF PROBLEM STRUCTURES

To demonstrate the reliability of our problem-specific structural classification, we convert categorical assignments into one-hot encodings. For each instance, we compute the mean and variance of each category's indicator across constraints and variables. This results in a twelve-dimensional feature vector: six dimensions for constraints (three category proportions and three variances) and six dimensions for variables (analogous calculations). We then apply principal component analysis (PCA) (Abdi & Williams, 2010) to these twelve-dimensional descriptors, projecting them onto a two-dimensional plane.

As shown in Figure 3, instances from the same problem family form tight clusters, while different families are well-separated. This confirms that our block-based structural fingerprint effectively characterizes MILP problem identity.

### A.5 ADDITIONAL RESULTS

**Structural feature design.** We conducted extensive experiments to incorporate structural information into the bipartite graph representation, exploring the following approaches:

- **Dec1:** Adding variable types and constraint types to the node features.
- **Dec2:** Adding variable types and constraint types to the node features, and incorporating (constraint, variable) pairs into the edge features.
- **Dec3:** Incorporating (constraint, variable) pairs into the edge features only.

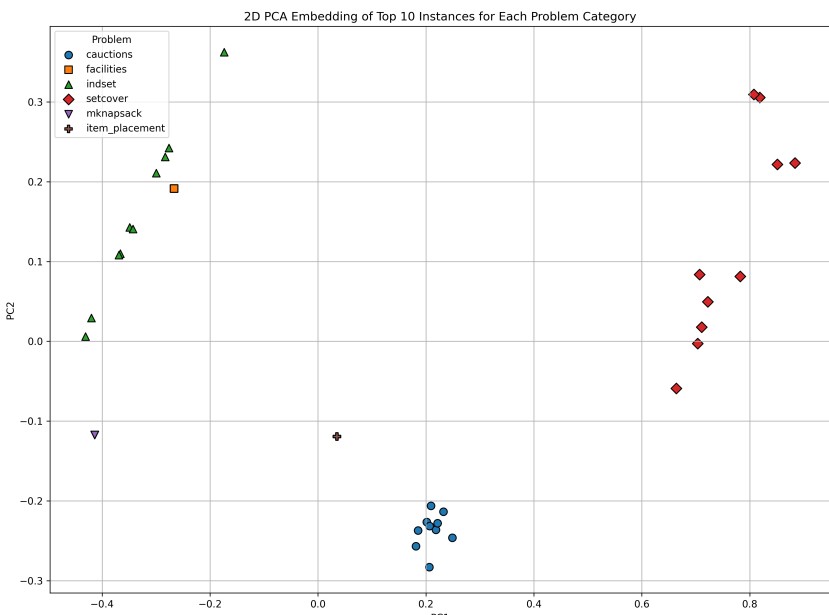

Figure 3: Principal Component Analysis of Block-Structure Features Across Six MILP Families.

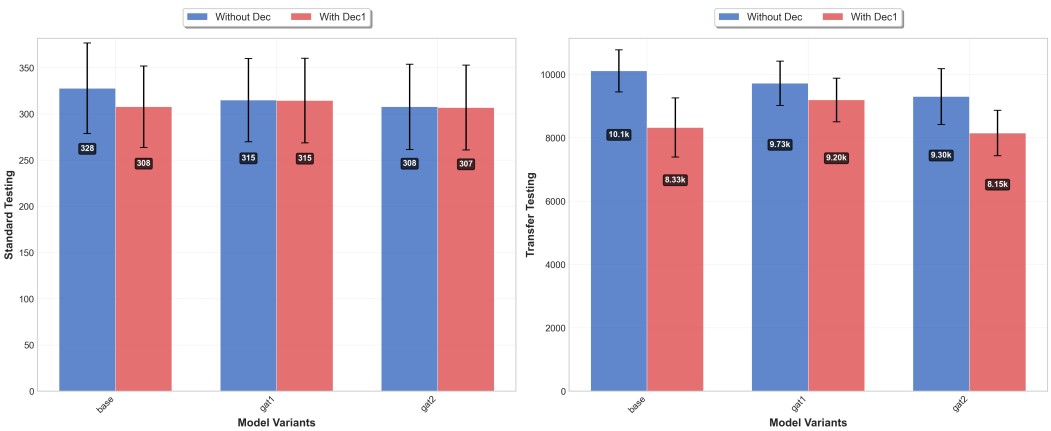

Figure 4: Dec1 Result in Cauctions Standard and Transfer Testing

- **Dec4:** Adding variable types and constraint types to the node features, and incorporating (constraint, variable) pairs into the node features.
- **Dec5:** Adding variable types and constraint types to the variable features, and incorporating (constraint, variable) pairs into both the node and edge features.

We evaluate these designs across three distinct network architectures: the vanilla GNN, GAT, and GAT2 (both attention-based models), utilizing the *Cauctions* dataset. The experimental results are presented in Figures 4–8 and analyzed as follows.

**Dec1** exhibits the most stable and substantial improvements across all model architectures and both evaluation scenarios. For the **transfer testing instances**, all three models demonstrate consistent performance gains, achieving an average improvement of **+11.83%** with a maximum of **+17.69%**. For the **standard testing instances**, while the improvements are more modest, they remain consistently positive across all models, with an average gain of **+2.16%**. These results demonstrate that augmenting variable features with variable and constraint type information effectively enables models to capture structural patterns while preserving robust generalization capabilities.

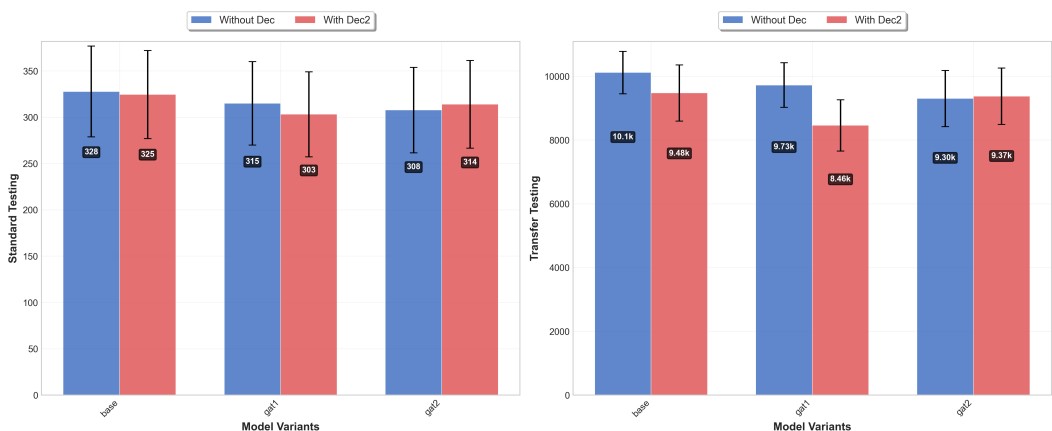

Figure 5: Dec2 Result in Cauctions Standard and Transfer Testing

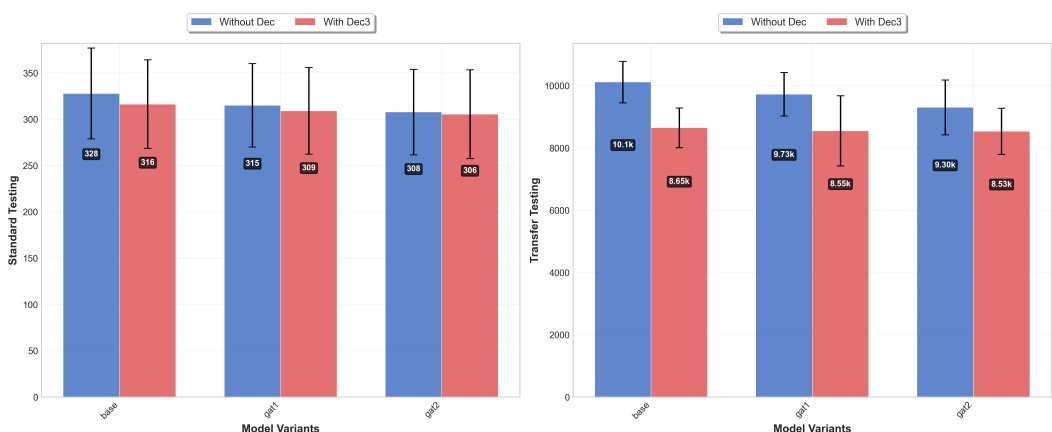

Figure 6: Dec3 Result in Cauctions Standard and Transfer Testing

**Dec2** demonstrates heterogeneous performance across different model architectures. On the **transfer testing instances**, the vanilla GNN and GAT models achieve notable improvements (up to **+13.00%**), while GAT2 experiences a marginal decline, yielding an overall average improvement of **+6.18%**. For the **standard testing instances**, the performance gains are limited (average **+0.91%**). This suggests that simultaneously enriching both variable and edge features with structural information provides moderate benefits but may introduce redundancy or noise, particularly for more sophisticated attention-based architectures.

**Dec3** achieves performance comparable to Dec and represents the second most stable design variant. For the **transfer testing instances**, all three models exhibit consistent improvements, with an average gain of **+11.63%** and a maximum of **+14.54%**. On the **standard testing instances**, the improvements remain modest yet consistently positive (average **+2.04%**). These results indicate that incorporating (constraint, variable) pairs exclusively into edge features effectively enriches relational information without introducing excessive representational complexity.

**Dec4** displays highly variable performance characteristics across model architectures. For the vanilla GNN, performance deteriorates substantially on the **transfer testing instances** ($-7.98\%$), whereas GAT and GAT2 achieve considerable improvements ($+16.09\%$ and $+7.86\%$, respectively). These findings suggest that this design, which integrates both type information and (constraint, variable) pair features into variable representations, necessitates sufficient model capacity to effectively utilize the enriched information. Simpler architectures may be overwhelmed by the increased complexity, resulting in performance degradation.

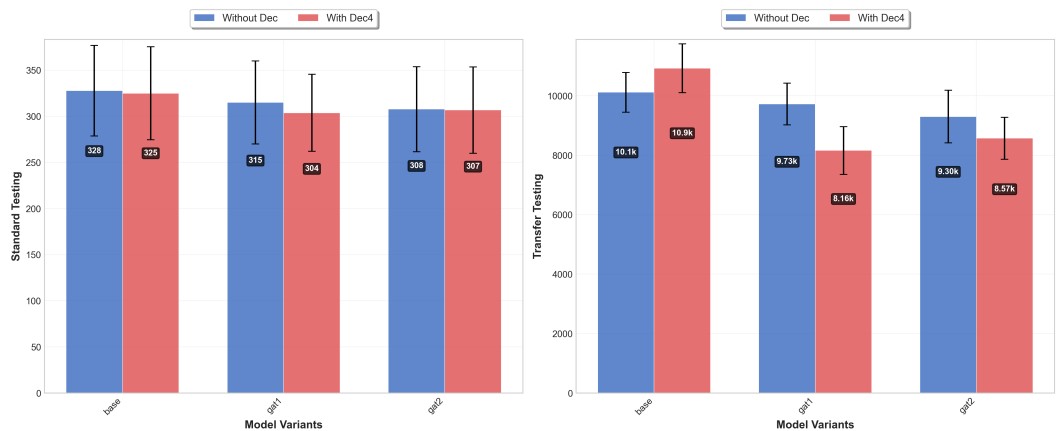

Figure 7: Dec4 Result in Cauctions Standard and Transfer Testing

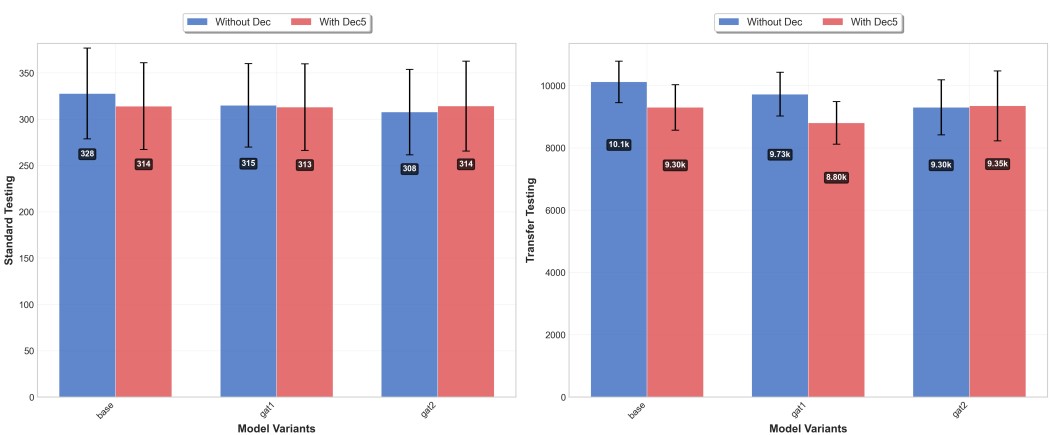

Figure 8: Dec5 Result in Cauctions Standard and Transfer Testing

**Dec5** exhibits limited and inconsistent improvements, similar to Dec2. For the **transfer testing instances**, the average improvement is merely **+5.70%**, with GAT2 again experiencing a slight performance decline. On the **standard testing instances**, the effect is even more attenuated (average **+0.89%**). These results indicate that incorporating comprehensive structural information into both variable and edge features introduces excessive redundancy, exceeding the models' learning capacity and ultimately diminishing effectiveness.

## A.6  LLM USAGE

Large Language Models (LLMs) were used solely for language refinement, including grammar checking, sentence rephrasing, and improving clarity and readability. The LLM had no involvement in the ideation, research methodology, experimental design, or data analysis.

All scientific content and conclusions were entirely developed by the authors, who take full responsibility for the manuscript. The use of the LLM adhered to ethical standards and did not result in plagiarism or scientific misconduct.

