# OpenReview forum: "Structure-Aware Bipartite Representations for Efficient MILP Branching"
_ICLR.cc/2026/Conference — ICLR 2026 Conference Withdrawn Submission_

### Official Review · Reviewer_qFb4 · 2025-10-24

**Soundness:** 2
**Presentation:** 3
**Contribution:** 2
**Rating:** 2
**Confidence:** 5

**Summary:**

The paper augments the standard bipartite graph representation for MILPs with block-structure annotations: one-hot role indicators for variables/constraints (e.g., master/block/border) and a normalized block-ID feature on edges to encode locality and inter-block couplings. The author trains a GNN-based branching policy via imitation learning of strong branching and integrates the learnt branching policy into SCIP within the Branch-and-Bound framework.

**Strengths:**

The authors propose leveraging MILP block structure to augment learn-to-optimize methods, highlighting the potential of how problem-specific information can improve L2O efficiency.

**Weaknesses:**

1. Integrating block information directly into the graph may be redundant. In the adopted Learn-to-branch framework, GNN message passing aggregates global context to each node and could already infer block structure; moreover, alternatives like graph pooling (e.g., [1]) offer a principled way to capture such coarse structure.
2. The numerical results do not demonstrate clear practical gains: the method outperforms FSB in some settings but is slower than PB/RPB on most datasets, and general purpose solvers like SCIP typically use hybrid branching strategies that dynamically combines multiple branching policies, which can be more efficient than relying on a single branching.
3. Relying entirely on GNNs for prediction may be inefficient: as graph size grows, GNN computation scales poorly comparing to conventional algorithms ([2]), and can become much slower than well-developed strategies like RPB.


[1] Grattarola, Daniele, et al. "Understanding pooling in graph neural networks." IEEE transactions on neural networks and learning systems 35.2 (2022): 2708-2718.
[2] Gupta, Prateek, et al. "Hybrid models for learning to branch." Advances in neural information processing systems 33 (2020): 18087-18097.

**Questions:**

1. Some symbols are not introduced; e.g., $z_t$ in the dual-integral computation is unspecified. Please define all variables when first used.
2. For minimization problems like Item Placement and Load Balancing, the dual bound $z_t$ should be a lower bound, and the dual integral should be Tx-F_0^T z_t dt, which is to be minimized. Why is this metric different from the original work?
3. In Item Placement, branching first on those 5 significantly large items can drastically cut solve time. Is it possible for the author to report the first few branching decisions under FSB, RPB, and the proposed method.
4. Has the author considered augmenting intra-block patterns with additional edges or hyper-nodes? If so, please justify why the current augmentation suffices (or provide results for the extended variant).

---

### Official Review · Reviewer_wrMG · 2025-10-26

**Soundness:** 3
**Presentation:** 3
**Contribution:** 2
**Rating:** 4
**Confidence:** 5

**Summary:**

In this article, the authors present a method to encode MILP instances in a pertinent manner, using block structure of the matrix constraints.

The main contributions are:

- Proposition of a clear methodology to encode the instance into a graph structure

- Extensive experimental comparison with existing methods

- The results show -- to a certain extent -- that the authors' methodology is effective

**Strengths:**

The article addresses a relevant problem in the MILP community: the encoding of the instances used for data driven approaches using machine learning.  The authors use 6 different types of instance in their experiments. This is valuable as performance of data driven techniques in MILP solving is known to heavily depend on the type of instance.

**Weaknesses:**

Despite extensive experiments, the overall contribution feels limited.
It would have been interesting to reposition the paper in the broader question of how to encode MILPs for data driven techniques. The focus of the paper is limited to GNNs, which is a popular method, but it could be the case that other encodings, even with simple neural networks, would be as effective.

Also, I find the results shown in the table intriguing/disappointing: please see my first question below.

**Questions:**

- In Table 2 the bolded lines are only those of the GNNs, but it appears that other baselines have better running times for example. For example, for (MK), the average running time with the baseline RPB is 19.46 and with the GNNs, 93.59, which is much more. Could the authors clarify why the GNN take so much time compared to the baseline?

- I suppose that the arrow (up/down) next to the type of metric (node, gap, time, ...) means: if up, then the greater the better, if down, the lower the better. If this is the case, I find it to be superfluous information. If not, please clarify the meaning?

---

### Official Review · Reviewer_rJ1U · 2025-10-28

**Soundness:** 2
**Presentation:** 3
**Contribution:** 2
**Rating:** 2
**Confidence:** 3

**Summary:**

This paper proposes a method that exploits the block diagonal structure of the constraint coefficient matrix (CMM) in GNN-based MILP solvers. The authors categorize constraint and variable nodes into three types, using these categories as augmented node features. Additionally, edges are labeled with block indices to incorporate structural information. This approach is integrated into Gasse et al.'s (2019) imitation learning framework for branching. Experimental results demonstrate that the proposed method is competitive with existing approaches.

**Strengths:**

- The proposed method is simple yet effective in leveraging the block structure of CMMs.
- The method is clearly described and easy to follow.

**Weaknesses:**

- The presentation of the block structure in CMMs could be improved for better clarity （The picture）.
-The experimental evaluation lacks comprehensiveness. Some important details regarding the baseline methods are missing. Additionally, the use of dual integral as a metric may be problematic (see Q3 below).

**Questions:**

1. **Table 1 appears incomplete:** The text mentions seven datasets, but the table does not seem to reflect this. Please verify and update accordingly.
2. **Details on baseline methods:** Could you clarify the implementation details of GNN, GNN_DEC, and GNN_DEC2? Specifically, are random features used in these baselines? It is also unclear how much the augmented features contribute to performance.
3. **Comparison with SOTA:** Have you compared your method with recent state-of-the-art approaches? For instance, a symmetry-aware GNN-based MILP solver has been proposed recently. A comparison with such methods would strengthen the contribution.
4. **Dual integral metric:** The definition of the dual integral reward appears inconsistent with Gasse et al. (2022). Please double-check for consistency. Additionally, the solution quality on the Item Placement and Load Balance datasets is not well presented—please provide more detailed results or analysis.

---

### Official Review · Reviewer_LNne · 2025-10-28

**Soundness:** 2
**Presentation:** 3
**Contribution:** 2
**Rating:** 2
**Confidence:** 4

**Summary:**

This manuscript proposes a feature-augmentation scheme that leverages the block-structure information of the constraint matrix to improve model performance in MILP branching. The key components include the categorization and identification of three types of block structures, and the injection of corresponding features into a bipartite representation. While the paper is generally well written and the proposed idea is interesting, the motivation is not sufficiently articulated and the experimental comparisons with existing methods are limited.

**Strengths:**

1. The manuscript is well-organized and follows a logical flow.
2. The idea of exploiting block-structure information in MILPs for improving learning-based branching is interesting and potentially impactful.
3. The empirical results demonstrate improvement over the chosen baselines.

**Weaknesses:**

#### Major weaknesses:

1. **Limited discussion of related feature-augmentation methods**: Several existing methods also perform feature augmentation to enhance model performance in MILPs. For instance, [1] introduces random noise, and [2] augments features based on symmetry orbits. Although these methods are motivated from different perspectives, they are still relevant to MILPs and should be discussed. Notably, the IP datasets used in this manuscript overlap with those in [2], making a comparative discussion particularly important for completeness.
2. **Insufficient and outdated learning-based baselines**: While Gasse et al. (2019) is indeed a seminal work, the field has progressed a lot in recent years. Recent methods such as [3] and [4] provide stronger baselines and more advanced architectures. Including at least one or two recent approaches would be essential to fairly assess the contribution of the proposed method.
3. **Unclear motivation and intuition**: Although the manuscript presents certain block structures observed in MILPs, it does not clearly explain **why** incorporating such structures would improve branching predictions. A clearer discussion of the connection between block-structure properties and branching decision quality would significantly strengthen the motivation.


#### Minor weaknesses:

1. Inconsistent dimensionality of $x_i$—it is sometimes treated as a scalar and other times as a vector. Please ensure consistent dimensionality throughout. For clarity and readability, it is recommended to use a conventional notation scheme: bold uppercase for matrices, bold lowercase for vectors, and regular (non-bold) fonts for scalars.
2. The term “structure-aware” is too general given that the manuscript only addresses block structures. It is recommended to revise the related wording to precisely reflect the scope so readers have the correct expectation.
3. The equation number for (1) appears misaligned or squeezed; please correct the formatting issue.


---

[1] Chen, Ziang, et al. "On Representing Mixed-Integer Linear Programs by Graph Neural Networks" 2022.

[2] Chen, Qian, et al. "When GNNs meet symmetry in ILPs: an orbit-based feature augmentation approach." ICLR 2025.

[3] Chen, Ziang, et al. "Rethinking the capacity of graph neural networks for branching strategy." NeurIPS 2024.

[4] Lin, Jiacheng, et al. "Cambranch: Contrastive learning with augmented milps for branching." ICLR 2024.

**Questions:**

1. What is the time complexity of identifying the block structures in a given constraint matrix?
2. In Figure 1, what exactly distinguishes the “Reorder” step from “Structure Detect”? Why does the constraint matrix after reordering appear more irregular?

---

### Official Review · Reviewer_pt4m · 2025-10-30

**Soundness:** 1
**Presentation:** 2
**Contribution:** 1
**Rating:** 2
**Confidence:** 4

**Summary:**

The goal of developing structure-aware GNNs for MILPs with block structures is commendable, but the execution in this work falls short. The proposed method offers limited novelty over existing GNN architectures, and its empirical validation is unconvincing. The performance improvements are marginal and fail to consistently surpass established baselines, undermining the claim of a significant advancement.

**Strengths:**

This work proposed to take advantage of the inherent block structures within MILPs. I think this is an interesting perspective.

**Weaknesses:**

I have several significant concerns regarding the methodological novelty, experimental design, and demonstrated efficacy of the proposed approach.

- Limited Methodological Novelty: The central claim of a novel representation for exploiting block structure is questionable. GNNs inherently capture localized structure, as their fundamental operation is based on message-passing between neighboring nodes. The authors must provide a clearer distinction, from a computational perspective, of how their proposed representation differs meaningfully from a standard GNN applied directly to a constraint-variable bipartite graph. Furthermore, the work would be significantly strengthened by theoretical justification, perhaps through the lens of representation theory, to formally characterize the representational power or efficiency gained by their proposed architecture over existing methods.

- Inappropriate Benchmark Selection: The empirical evaluation relies on standard combinatorial problems like SC and MIS, which typically do not exhibit the pronounced block structure this method aims to exploit. This choice undermines the core premise of the work. A more compelling and appropriate validation would involve problems with explicit, known block structures, such as Two-stage Stochastic MILPs, which would directly test the method's advantages.

- Marginal Performance Gains: The reported performance improvements over strong baselines (including GNNs and classical branching rules) are minor and not consistently superior. In several instances, the proposed method fails to outperform these baselines. Without a clear and statistically significant performance advantage, the practical utility of the proposed representation remains unproven.

**Questions:**

See the weaknesses.

---

### Note · Authors · 2025-12-02

I have read and agree with the venue's withdrawal policy on behalf of myself and my co-authors.